# Jahn-Teller-induced femtosecond electronic depolarization dynamics of the nitrogen-vacancy defect in diamond

Ronald Ulbricht[1,2,†], Shuo Dong[2], I-Ya Chang[3,4], Bala Murali Krishna Mariserla[5], Keshav M. Dani[5], Kim Hyeon-Deuk[3,4] & Zhi-Heng Loh[1,2,6]

Single-photon emission from the nitrogen-vacancy defect in diamond constitutes one of its many proposed applications. Owing to its doubly degenerate $^3E$ electronic excited state, photons from this defect can be emitted by two optical transitions with perpendicular polarization. Previous measurements have indicated that orbital-selective photoexcitation does not, however, yield photoluminescence with well-defined polarizations, thus hinting at orbital-averaging dynamics even at cryogenic temperatures. Here we employ femtosecond polarization anisotropy spectroscopy to investigate the ultrafast electronic dynamics of the $^3E$ state. We observe subpicosecond electronic dephasing dynamics even at cryogenic temperatures, up to five orders of magnitude faster than dephasing rates suggested by previous frequency- and time-domain measurements. *Ab initio* molecular dynamics simulations assign the ultrafast depolarization dynamics to nonadiabatic transitions and phonon-induced electronic dephasing between the two components of the $^3E$ state. Our results provide an explanation for the ultrafast orbital averaging that exists even at cryogenic temperatures.

[1] Division of Chemistry and Biological Chemistry, School of Physical and Mathematical Sciences, Nanyang Technological University, Singapore 637371, Singapore. [2] Division of Physics and Applied Physics, School of Physical and Mathematical Sciences, Nanyang Technological University, Singapore 637371, Singapore. [3] Department of Chemistry, Kyoto University, Kyoto 606-8502, Japan. [4] Japan Science and Technology Agency, PRESTO, 4-1-8 Honcho, Kawaguchi, Saitama 332-0012, Japan. [5] Femtosecond Spectroscopy Unit, Okinawa Institute of Science and Technology Graduate University, 1919-1 Tancha, Onna-son, Kunigami, Okinawa 904-0495, Japan. [6] Centre for Optical Fibre Technology, The Photonics Institute, Nanyang Technological University, Singapore 639798, Singapore. † Present address: Department of Physics, Department of Chemistry and JILA, University of Colorado, Boulder, Colorado 80309, USA. Correspondence and requests for materials should be addressed to R.U. (email: ronald.ulbricht@colorado.edu) or to K.H.-D. (email: kim@kuchem.kyoto-u.ac.jp) or to Z.-H.L. (email: zhiheng@ntu.edu.sg).

Among the known colour centres of diamond, the negatively charged nitrogen-vacancy (NV$^-$) defect has attracted the most attention[1], motivated by its potential to serve as a building block for novel quantum technologies. Remarkable advances in their magnetic and optical manipulation, performed even at the single-defect level[2], herald their application to spin-based quantum computing[3,4] and photonics[5], as well as nanoscale magnetic field[6–8] and temperature sensors[9–11].

Buried deep within the band gap of diamond are the NV$^-$ $^3A_2$ electronic ground state and the doubly degenerate $^3E$ excited state, which are optically coupled by a narrow zero-phonon line (ZPL) transition at 1.95 eV (637 nm wavelength, Fig. 1a). This optical transition has been identified as a potential quantum emitter for single photons[1,5]. Vibronic coupling between the $^3E$ electronic state and a quasi-localized vibrational mode at $\sim68$ meV gives rise to a broad phonon sideband[12]. The $E_x$ and $E_y$ sublevels of the $^3E$ state have orthogonal electronic alignment, evidenced by the computed orbital densities in Fig. 1b, hence yielding two perpendicularly polarized $^3A_2 \to {}^3E$ transitions[1].

As a result of its orbital degeneracy, the $^3E$ state couples to a doubly degenerate vibrational mode of $e$ symmetry to form an $E \otimes e$ Jahn-Teller (JT) system[1]. The JT-active mode involves the displacement of the carbon atoms that surround the vacancy. The initially degenerate $E_x$ and $E_y$ states are displaced along the JT-active mode $Q_{JT}$, creating a conical intersection (CI, see Fig. 1b). At the same time, the potential minimum energy is reduced by the JT stabilization energy $E_{JT}$. For the NV$^-$ defect, the theoretically predicted JT stabilization energy $E_{JT}$ of 25 meV is smaller than the tunnelling splitting of 34 meV, therefore rendering the system a dynamic JT distortion[13].

In the vicinity of CIs, the Born-Oppenheimer approximation breaks down, allowing exceptionally fast nonadiabatic (NA) transitions between potential energy surfaces[14,15]. In molecular photochemistry, for example, CIs between electronic excited and ground states are known to promote ultrafast internal conversion on femtosecond timescales[16]. In the case of the NV$^-$ defect, the CI exists only between the excited state $E_x$ and $E_y$ orbitals, thereby potentially enabling ultrafast NA transitions between them. In addition, the JT effect has been invoked to explain the asymmetry between the absorption and photoluminescence (PL) lineshapes[1], the broadening of the $^3E \to {}^3A_2$ ZPL transition linewidth with increasing temperature and the concomitant reduction in the polarization contrast of its PL[17]. These JT effects have important ramifications. The non-unity polarization contrast, observed even at cryogenic temperatures, for example, impairs coupling of NV$^-$ photon sources to plasmonic waveguides and decreases the interference visibility of emitted single photons, thus impeding its use in quantum information processing[5].

The coherent dynamics of the orbital doublet at the $E \otimes e$ JT CI are encoded in the optical dephasing times. From the PL excitation linewidths of single defects, dephasing times of $\sim10$ ns at sub-10 K and $<0.3$ ps at temperatures beyond 200 K have been inferred[17]. The former is substantiated by an orbital coherence time of $\sim6$–7 ns, directly determined from time-domain Ramsey fringe interferometry[18]. However, the low PL polarization contrast of $\lesssim0.5$ at 4 K indicates the existence of ultrafast dynamics that precede the nanosecond electronic dephasing processes that have been uncovered so far[17–19].

Here we use femtosecond polarization anisotropy (PA) spectroscopy to resolve the ultrafast coherent orbital dynamics

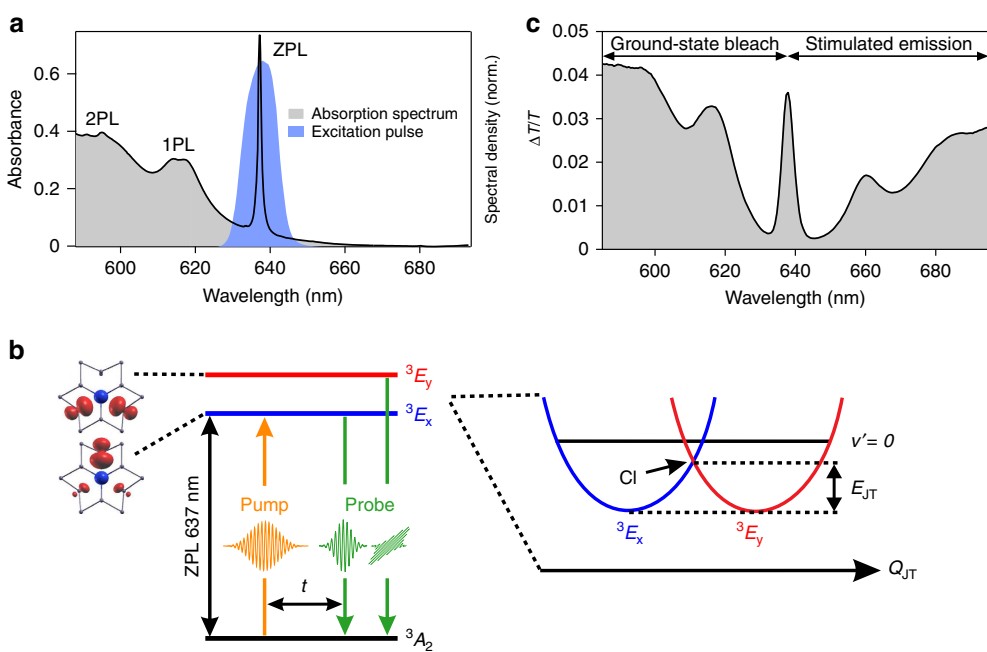

**Figure 1 | Electronic and optical properties of the NV$^-$ defect and the polarization anisotropy spectroscopy measurement scheme. (a)** Optical absorption spectrum of the NV$^-$ sample recorded at 77 K, showing a distinct ZPL, in addition to the one- (1PL) and two-phonon line (2PL) transitions. The spectrum of the pump pulse used to excite the ZPL line is shown in blue. **(b)** Electronic structure of the NV$^-$ defect, showing the $E_x$ (blue) and $E_y$ (red) components of the $^3E$ state accessed from the $^3A_2$ ground state via the ZPL transition at 1.95 eV (637 nm). Structural distortion along a JT-active mode $Q_{JT}$ lowers the symmetry of the doubly degenerate $^3E$ state, found at the CI, to yield JT-split $^3E_x$ and $^3E_y$ states. The JT stabilization energy $E_{JT}$ of 25 meV positions the CI below the $v' = 0$ level, located at 34 meV. In the experiments, a linearly polarized pump excites the $^3E$ doublet and a time-delayed broadband probe pulse with variable polarization, either parallel or perpendicular relative to the pump, interrogates the time-resolved stimulated emission. The orbital plots show the orbital alignment for the $E_x$ and $E_y$ states, as viewed along the $C_3$ axis of the NV$^-$ defect. The red regions denote the electron probability distribution and the blue sphere denotes the nitrogen atom. The plots are calculated by density functional theory using the PBE functional and projector-augmented-wave pseudopotentials (see Methods). **(c)** Spectrally resolved differential transmission $\Delta T/T$, showing contributions from ground-state bleaching (blue side of ZPL) and stimulated emission (red side of ZPL).

of the $^3E$ state. We observe biphasic electronic dephasing dynamics occurring on subpicosecond to few-picosecond timescales, even at cryogenic temperatures, up to five orders of magnitude faster than dephasing rates suggested by previous frequency- and time-domain measurements[17,19,20]. Ab initio molecular dynamics (AIMD) simulations assign the dynamics to NA transitions around the CI and phonon-induced electronic dephasing between components of the $^3E$ state.

## Results

**PA spectroscopy.** The femtosecond PA measurements employ a narrowband linearly polarized pump pulse to excite the ZPL transition of the NV$^-$ defect (Fig. 1a), following which a broadband linearly polarized probe pulse measures the pump-induced change of the normalized transmission spectrum $\Delta T/T$. Excitation of the ZPL as opposed to the phonon sideband avoids complications introduced by ultrafast vibrational relaxation[21]. Varying the pump–probe time delay and relative polarization yields the time-resolved $\Delta T/T$ signal (Fig. 1b) for parallel ($S^{\parallel}$) and perpendicular ($S^{\perp}$) relative polarization between pump and probe pulses. The PA signal $S_{\text{aniso}}(\lambda,t)$ is then obtained from the relation $S_{\text{aniso}}(\lambda,t) = (S^{\parallel} - S^{\perp})/(S^{\parallel} + 2S^{\perp})$ (see Methods).

Photoexcitation leads to increased transmission of the NV$^-$ sample, as can be seen from the positive $\Delta T/T$ signal over the entire probe spectrum (Fig. 1c). Features on the blue side of the ZPL arise from depletion of the $^3A_2$ ground state by the photoexcitation pump pulse, resulting in the bleaching of the $^3A_2$ ground-state absorption spectrum. The positive $\Delta T/T$ signal on the red side of the ZPL is because of Stokes-shifted stimulated emission from the $\nu' = 0$ level of the $^3E$ state, populated by the pump pulse, to the various $\nu''$ levels on the $^3A_2$ ground state. As such, the former signal is sensitive to ground-state dynamics, whereas the latter is sensitive to excited state dynamics. Note that excited-state absorption from the $^3E$ state, which would give negative $\Delta T/T$ signals, is negligible because of the small oscillator strength of excited-state absorption into the conduction band.

$S_{\text{aniso}}(\lambda,t)$ provides information on the alignment dynamics after photoexcitation (see Methods and Supporting Information). In molecular spectroscopy, for instance, this is used to measure the reorientation of molecules in solution[22], an effect that does not occur here because the NV$^-$ defects are fixed in the diamond lattice. In addition, for probe transitions involving doubly degenerate excited states with perpendicular transition dipoles, as is the case here, the PA signals also reflect electronic reorientation. In such instances, the PA signal reports on electron motion around a CI[23,24], and its decay yields the dephasing time between the $E_x$ and $E_y$ states[25]. Since the measurements are performed on an ensemble of differently oriented NV$^-$ centres, linearly polarized photoexcitation does not selectively populate only one component of the orbital doublet, but instead, prepares a coherent superposition of the $E_x$ and $E_y$ states. As such, both population transfer and loss of phase coherence between the orbital doublet states—collectively referred to as orbital dephasing herein—lead to electronic de-alignment and, hence, the decay of the PA[23,25]. The observation of dephasing dynamics in the time-domain complements frequency-domain measurements, particularly when dephasing is fast and spans multiple timescales, which make the broad linewidths associated with ultrafast depolarization challenging to discern in the frequency domain.

The PA spectrum $S_{\text{aniso}}(\lambda,t)$ collected as a function of pump–probe time delay at 77 K is shown in Fig. 2a. The anisotropy value obtained on the blue side of the ZPL transition, where ground-state bleaching dominates, is found to be constant to within experimental error ($S_{\text{aniso}} \sim 0.1$). On the other hand, a

pronounced decay (Fig. 2b) is observed for the PA recorded at the ZPL (white-dashed line) and to its red side, where stimulated emission occurs. Anisotropy dynamics only appearing through stimulated emission indicates that they originate from the $^3E$ excited state. The PA decay at the ZPL can be fit to the function $S_{\text{aniso}}(t) = A_0 + A_1 e^{-t/\tau_1} + A_2 e^{-t/\tau_2}$, where the offset $A_0$, amplitudes $A_1$ and $A_2$, and exponential decay constants $\tau_1$ and $\tau_2$ are all fit parameters. The time constants of both the fast ($\tau_1$) and slow ($\tau_2$) decay components exhibit distinct temperature dependencies (Fig. 2c). Furthermore, $\tau_2$ is evident only below 150 K. $\tau_1$ is constant to within experimental error, varying between $0.15 \pm 0.03$ and $0.10 \pm 0.02$ ps for all measured temperatures. On the other hand, $\tau_2$ decreases markedly from $14.4 \pm 1.7$ ps at 10 K to $0.70 \pm 0.11$ ps at 150 K. It is noteworthy that $S_{\text{aniso}}$ does not completely decay to 0.10 within the maximum time delay of 20 ps employed in our measurements. This asymptotic value of 0.10 for $S_{\text{aniso}}$ would be expected for an electronic system that comprises a nondegenerate ground state and a doubly degenerate excited state with orthogonal transition dipoles, as is the case here[23,25]. We note that $S_{\text{aniso}}$ does not vanish to zero, as it would for molecules in solution by rotational diffusion[22] because the defects are fixed in the diamond lattice. We believe that the deviation of the final $S_{\text{aniso}}$ value from the theoretical isotropic value of 0.10 is indicative of long-lived orbital coherence surviving beyond 20 ps, consistent with nanosecond dephasing dynamics that have been observed at cryogenic temperatures via single-defect lineshape measurements[17] and time-domain Ramsey fringe interferometry[18]. The subpicosecond to picosecond dynamics observed herein precede the previously reported dephasing timescales.

**Subpicosecond $\tau_1$ electronic depolarization dynamics.** NA AIMD simulations are performed to elucidate the origin of the observed subpicosecond depolarization dynamics. The computed decay profiles reveal depolarization via population transfer from the initially populated $E_x$ state to the $E_y$ state (Fig. 2d). The $\sim 100$-fs timescale for electronic equilibration (Fig. 2e) is in agreement with the experimental $\tau_1$ values. The inverse relation between the computed $\tau_1$ and the electron–phonon coupling strength (Fig. 2e) indicates that the fast component of the electronic depolarization dynamics is driven by NA transitions. The relatively weak temperature dependence is partly due to the stiff and extended diamond structure, and is qualitatively different from the strong temperature dependence that appears, for example, in semiconductor quantum dots, which have finite nanoscale size and thus a large surface-to-volume ratio[26]. Fourier transforms of the real-time fluctuations of the $E_x$ and $E_y$ energies yield the frequencies of the phonon modes that mediate the electronic equilibration (Fig. 2f). Within the theoretical framework of AIMD, these phonon-induced fluctuations arise from incoherent vibrational motions of the thermal bath (see Supplementary Figs 1 and 2), unlike coherent phonons launched by impulsive excitation, which manifest themselves as oscillatory features in the $\Delta T/T$ signals. The fast fourier transform (FFT) power spectrum obtained at 77 K reveals several prominent peaks at 47, 69, 90, 120, 130 and 150–160 meV. These modes have been identified and assigned by earlier ab initio calculations to the various quasi-localized vibrational modes of the NV$^-$ defect[27], including one at 69 meV that coincides with the energy of the JT-active $e$ mode[13]. Interestingly, the presence of multiple phonon frequencies in the FFT power spectrum suggests that the depolarization dynamics are driven not only by the JT-active modes, but also by a collection of other vibrational modes that are anharmonically coupled to the JT-active modes.

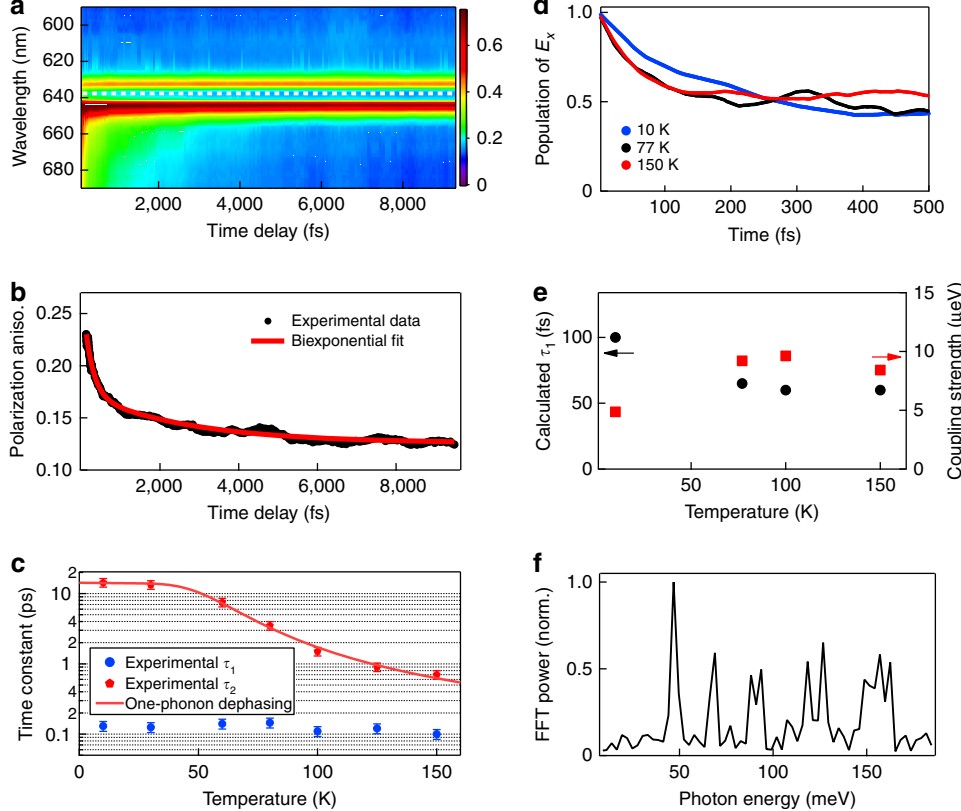

**Figure 2 | Ultrafast electronic depolarization dynamics and results of the NA AIMD simulations.** (**a**) Time evolution of the polarization anisotropy spectrum recorded at 77 K after ZPL excitation. (**b**) Line-out at the ZPL wavelength (white dashed line in **a**) showing a biexponential decay of the PA with time constants $\tau_1$ and $\tau_2$. (**c**) Temperature dependence of $\tau_1$ and $\tau_2$ and fit of the temperature dependence of $\tau_2$ to equation (1); error bars denote the s.e.m. (**d**) Simulated electronic depolarization trajectories for various temperatures. (**e**) Time constant $\tau_1$ and electron–phonon coupling strength computed at various temperatures. (**f**) Simulated FFT power spectrum at 77 K showing the frequencies of phonon modes that mediate ultrafast electronic depolarization. Aniso, anisotropy; norm, normalized.

**Picosecond $\tau_2$ electronic depolarization dynamics**. The conspicuous temperature dependence of $\tau_2$ suggests phonon-mediated electronic depolarization as its origin. The vanishing NV$^-$ defect phonon density of states[13,27,28] at phonon energies $<20$ meV prevents one-phonon transitions from effecting population transfer between the $E_x$ and $E_y$ states[29]. Nevertheless, one-phonon-mediated electronic depolarization is still possible via phonon-induced fluctuations of the orbital doublet, or even transitions from the lowest-energy $E_x$ and $E_y$ vibronic levels of the $^3E$ electronically excited state to higher-lying, nondegenerate vibronic levels of $A_1$ and/or $A_2$ symmetry, for which electronic alignment does not exist. In either case, the one-phonon transition rate $\Gamma(T) = 1/\tau_2(T)$ follows the relation[30]

$$\Gamma(T) = \Gamma_0 + \sigma T + \Gamma_c / \left( e^{\hbar\omega_c / k_B T} - 1 \right), \quad (1)$$

where $\Gamma_0$ is the temperature-independent offset, $\sigma$ ($\Gamma_c$) characterizes the electron–phonon coupling strength for phonon energies $\hbar\omega \ll k_B T$ ($\hbar\omega \gtrsim k_B T$) and $\omega_c$ represents the effective frequency of a group of phonons with energies $\hbar\omega \gtrsim k_B T$ that induces depolarization. Fitting the experimental data to equation (1) gives $\Gamma_0 = 0.070 \pm 0.013$ ps$^{-1}$, $\sigma \sim 0$ (to within experimental uncertainty), $\Gamma_c = 9.9 \pm 3.4$ ps$^{-1}$ and $\hbar\omega_c = 26 \pm 3$ meV (Fig. 2c). The vanishing $\sigma$ coefficient is consistent with the paucity of low-energy phonons with $\hbar\omega \ll k_B T$ in the phonon density of states. Interestingly, $\omega_c$ closely approaches the calculated $E{-}A_1$ tunnelling splitting[13] of 35 meV, suggesting possible electronic depolarization via the phonon-driven population transfer from the $E_x$ and $E_y$ vibronic

levels to the totally symmetric $A_1$ vibronic level within the $^3E$ electronically excited state. Excited-state AIMD simulations based on a microcanonical trajectory generated at 10 K furnish a dephasing time of 8 ps, in good agreement with the experimentally measured value of $14.4 \pm 1.7$ ps (see Supplementary Fig. 3). The correlated phonon-induced fluctuations of the $E_x$ and $E_y$ states support the long dephasing time. We note, however, that the experimental uncertainty in the measured $\tau_2$ values does not allow depolarization via one-phonon transitions to be distinguished from two-phonon Raman-type population transfer[17] between the $E_x$ and $E_y$ states (see Supplementary Fig. 4). The latter, whose rate scales as $T^5$, has been invoked to explain the temperature-dependent population transfer and electronic dephasing rates obtained from measurements of single-defect linewidths[17,31], decoherence of Rabi oscillations[29] and ensemble photon-echo spectroscopy[32].

## Discussion
Previous frequency- and time-domain measurements were performed on high-purity Type 2a diamond samples, which have defect densities that are in the parts-per-billion (p.p.b.) regime, and thus orders of magnitude lower than the Type 1b sample ($\sim 100$ p.p.m.) used in the present study. Therefore, a natural question that arises is the extent to which the observed ultrafast dynamics are intrinsic to a single defect. Fluctuations of the charge bath and defect–defect interactions could presumably lead to the enhanced dephasing rates observed herein. First, we note that previous single-defect measurements on samples with

**Figure 3 | Ultrafast electronic depolarization dynamics of the NV⁻ defect.** (**a**) Initial polarized photoexcitation leads to the population of the $E_x$ level at $t = 0$. (**b**) Partial electronic depolarization occurs via NA transitions between $E_x$ and $E_y$ on a timescale of $\tau_1$. (**c**) Various possible pathways that drive further electronic depolarization on a timescale of $\tau_2$ include phonon-induced fluctuations of the $E_x$ to $E_y$ levels (wavy arrows), resonant one-phonon excitation to the unaligned $A_1$ vibronic level (solid arrows) and off-resonant two-phonon population transfer between $E_x$ and $E_y$ (dashed arrows). For simplicity, initial excitation is assumed to populate only the $E_x$ state.

similarly high defect densities[33,34] yield microsecond timescales for spectral diffusion, orders of magnitude longer than the picosecond and subpicosecond dynamics observed here. Second, considering the quasi-localized nature of the vibrational modes[27], the modification of the phonon density of states of a given NV⁻ centre by an adjacent defect is expected to be negligible. Finally, and most importantly, we point out that the AIMD simulations were performed on a 215-atom supercell, which corresponds to an effective defect density of 4,650 p.p.m., $> 40 \times$ larger than the actual $\sim 100$ p.p.m. defect density of the sample used in the measurements. The fact that the observed dephasing timescales are reproduced by the AIMD simulations strongly suggests that the ultrafast dephasing dynamics are intrinsic to the isolated NV⁻ centre. This claim can be verified by future measurements on defects with varying NV⁻ densities.

Our combined experimental–theoretical investigation affords the following unified picture of the ultrafast electronic depolarization dynamics following ZPL photoexcitation of the $^3E$ excited state. Photoexcitation by a linearly polarized laser pulse at the ZPL creates a NV⁻ defect that is electronically aligned in the excited state along the polarization axis of the laser field, leading to a coherent superposition of the $E_x$ and $E_y$ states (Fig. 3a). Because of the energetic proximity of the vibronic levels to the JT CI, where the electron–phonon coupling strength is maximal, the non-vanishing phonon velocities of the heat bath promote efficient NA transitions between the $E_x$ and $E_y$ states, leading to rapid electronic depolarization on the $\tau_1 \sim 0.1$-ps timescale (Fig. 3b). On longer timescales spanning $\tau_2 \sim 1 - 10$ ps, orbital dephasing is promoted by electron–phonon scattering involving phonon-induced fluctuations, transitions to totally symmetric, higher-energy vibronic levels and/or two-phonon population transfer between the $E_x$ and $E_y$ states (Fig. 3c).

The subpicosecond to picosecond electronic depolarization dynamics unravelled here provide an explanation for the hitherto unaccounted loss of polarization fidelity of the NV⁻ defect PL at cryogenic temperatures[17,19], that is, the fact that orbital averaging occurs even at such low temperatures. This ultrafast biphasic dephasing could have eluded previous frequency-domain measurements since, insofar as multiple disparate dephasing timescales are involved, it is conceivably challenging to identify in a lineshape analysis the broad pedestal that is associated with subpicosecond dephasing. Note that time-domain photon echo measurements have elucidated similar biphasic dephasing dynamics spanning two to three orders of magnitude for excitons in self-assembled[35] and colloidal[36] quantum dots. In the case of the NV⁻ centre, our results, together with earlier frequency[17,19] and time-domain studies[18,32], demonstrate that electronic dephasing at a given cryogenic temperature spans an unprecedented five decades in time, from $10^{-13}$ to $10^{-8}$ s.

## Methods

**Sample.** The investigated NV⁻ sample is a high-pressure high-temperature-grown Type 1b diamond (Element Six) measuring $4 \times 4 \times 0.3$ mm³. NV⁻ defects are introduced by irradiation with 1-MeV electrons at a flux of $10^{18}$ cm⁻² and subsequent annealing in vacuum for 2 h at a temperature of 800 °C. The resulting NV⁻ density is $\sim 10$ p.p.m., whereas the density of remaining $N_s$ defects is 100 p.p.m. The pure diamond used for artefact subtraction is a CVD-grown type IIa diamond of similar size (Element Six).

**Ultrafast polarization-resolved optical spectroscopy.** Femtosecond PA measurements were performed on two ultrafast transient absorption set-ups: a broadband set-up that furnishes probe pulses spanning 550–750 nm and wavelength-tunable narrowband pump–pulses (10-nm bandwidth), and a two-colour set-up that uses tunable narrowband pulses (10-nm bandwidth) for both the pump and probe. The narrow pump-pulse bandwidth of both set-ups allows the selective excitation of the ZPL transition at 637 nm, after which the linearly polarized probe pulse measures the pump-induced change of the normalized transmission spectrum $\Delta T/T$. The time resolution is 80 fs (Supplementary Fig. 5). Signals for parallel ($S^\parallel$) and perpendicular ($S^\perp$) relative polarization between pump and probe pulses are recorded. The PA signal $S_{\mathrm{aniso}}(\lambda, t)$ is then obtained from the relation $S_{\mathrm{aniso}}(\lambda, t) = (S^\parallel - S^\perp)/(S^\parallel + 2S^\perp)$. The typical pump fluence is $\sim 1$ mJ cm⁻², which yields a $\Delta T/T$ signal of 0.036 at the ZPL transition wavelength (Fig. 1c). Such a small $\Delta T/T$ value confirms that our measurements are performed in the weak-perturbative limit and higher-order contributions to the signal are negligible. The sample was mounted in either a liquid-nitrogen-cooled cryostat (broadband set-up, 77–400 K) or a closed-cycle helium-cooled cryostat (two-colour set-up, 10–300 K). Further details on the experimental set-up, as well as data processing and analysis procedures, can be found in the Supplementary Methods and Supplementary Figs 4–8.

**AIMD simulations.** We performed real-time atomistic simulations for anisotropy decay of an initially created dipole moment in the NV⁻ defect. The electronic structures of both ground and excited states as well as their state-specific MD simulations were obtained with the VASP software package using the Perdew-Burke-Ernzerhof (PBE) density functional and projector-augmented-wave pseudopotentials. The geometry of the NV⁻ defect comprises 1 nitrogen atom and 214 carbon atoms with an additional single electron to achieve the negatively charged NV⁻ centre. The NV⁻ defect was heated up to various temperatures ranging from 10 to 300 K by repeated velocity rescaling, and a 5-ps microcanonical trajectory at each temperature was calculated on the ground and excited $E_x$ states using the Verlet algorithm with a 1-fs time step. In the excited state, we forced a spin-down electron to be located at the $\bar{e}_x$ orbital and removed symmetry constraints of the geometry. Real-time simulations for the PA decay were performed with the time-dependent NA electron–phonon coupling and orbital energies updated at every time step. Further details on the simulations can be found in the Supplementary Methods and Supplementary Figs 1–3,9 and 10.

**Data availability.** The data that support the findings of this study are available from the corresponding authors upon reasonable request.

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

## Acknowledgements

This work is supported by a start-up grant from NTU, funding from the A*Star Science and Engineering Research Council (122-PSF-0011 and 122 360 0008) and the Ministry of Education (MOE2014-T2-2-052), and the award of a Nanyang Assistant Professorship to Z.-H.L. K.H.-D. thanks the financial support from JST (PRESTO) and Grant-in-Aids for Scientific Research from Japan Society for the Promotion of Science (KAKENHI), Grant No 15K05386. R.U. acknowledges support by a Rubicon Grant of the Netherlands Organization for Scientific Research (NWO). We are grateful to D.M. Jonas, O.V. Prezhdo, H. Köppel, A. Gali, W. Peters and M. Cho for useful discussions, to R.U.A. Khan, Z. Wang and C. Soci for experimental assistance and to J. Schwartz for providing the sample.

## Author contributions

R.U. and Z.-H.L. conceived and designed the project. R.U., S.D. and B.M.K.M. performed the experiments. R.U. and Z.-H.L. analysed the data. I.-Y.C. and K.H.-D. performed the simulations. R.U., K.H.-D. and Z.-H.L. wrote the manuscript, with input from all authors.

## Additional information

**Competing financial interests:** The authors declare no competing financial interests.

