## [Peer Review File · Nature Communications]

Reviewers' comments:

Reviewer #1 (Remarks to the Author):

The manuscript presents ultrafast polarization anisotropy measurements to report electronic dephasing dynamics spanning multiple decades of time associated with the electronic transitions of the nitrogen vacancy defect center in diamond. Additionally, molecular dynamics simulations have been performed to investigate the origin of the observed components.

The paper represents a study that is potentially of significant interest across multiple research communities with implications for future experiments and theoretical development. However, there are some issues with the paper that must be addressed prior to publication as outlined below.

The authors invoke a relaxation pathway involving a JT distortion induced conical intersection. While the dynamic JT distortion is certainly associated with non-adiabatic dynamics, the authors need to be cautious when using this terminology. The use of this term requires more explanation and support. For instance, it would be easy for a reader to mistakenly interpret the CI as connecting the excited state to the ground state directly - which would lead to the erroneous idea that relaxation to the ground state occurs on a very short timescale. Also, the term CI is mentioned only at the very beginning of the manuscript and briefly on page 7 associated with fig. 3. At the very least, there should be a more thorough explanation of the role of the CI as presented in fig. 3.

In general, the paper would also benefit from clearer explanations of the results. Although this situation is improved by the supplemental information, there are many aspects of the work presented in the paper that are not sufficiently supported or explained. Even given the length restrictions, this could be done without adding a large amount of text. The paper would be vastly improved by a sentence or two clarifying the main points, results and techniques used. For instance, the vibrational modes contributing to the signal are not listed in a comprehensive manner. Also, no experimental FFT showing the vibrational modes is presented. Only a simulated FFT is shown. This is an issue for claiming the participation of particular JT associated vibrational modes. In addition, the reasons for using polarization anisotropy should be more clearly outlined. This is an appropriate experimental technique to use in this case, however, the authors could do a better job of explaining the motivation to the reader.

The absorbance seems exceptionally high for an optical experiment. In a third order experiment, absorbances of higher than 0.3 are known to potentially cause issues with contributions from higher order signals. This is true even in third order phase matched directions such as that for the pump probe experiment presented here. This can be mitigated by using lower power pulses (sub 10 nJ as a general guideline) or doing a power dependence to show when higher orders contribute. However, the paper does not list the power per pulse or show a power dependence. Without that information, it is unclear if higher order signals are potentially contributing.

The experiments involved polarization anisotropy on a sample with randomly distributed nitrogen vacancy defect centers. In this case, pre-factors associated with a static rotational average should have been considered.

Regarding format, the paper needs a conclusion. In the current form, it seems to just end without a statement of significance or a summary of the most important results.

Reviewer #2 (Remarks to the Author):

The article by Ulbricht et al. describes a combined experimental and theoretical investigation of picosecond timescale orbital dynamics within the electronic excited states of the diamond NV center. The methods appear sound (though need to be described a bit more clearly) and the experiments, simulations, and analysis appear to have been performed with a high level of rigor. I am not an expert on the ab initio simulations, so my comments are focused on the experimental and basic theoretical portions which comprise the majority of the manuscript. All in all I recommend that the paper eventually be published in a suitable journal after addressing my comments.

I have several comments on the manuscript which fall under the following categories:

1. The methods are not adequately described in the text and there is a reliance on the reader understanding the ultrafast optics jargon.
2. The sample used is a high-defect density $[N] \sim 100$ ppm sample. These samples have NV centers whose properties at low temperature are dramatically different from the single centers studied in nearly all of the previous works the authors cite. This casts a doubt on the relevance of this work. For example, is it clear the authors are studying processes intrinsic to NV centers and not related to interaction with a rapidly fluctuating charge bath or highly modified phononic environment or some other defect-defect interaction?
3. While the experimental techniques are impressive, the topic is a bit esoteric. It's hard to imagine this work impacting how researchers use NV centers in applications. The techniques themselves may be novel, but I am not qualified to judge.

Specific comments below:

A. Line 19: Not clear what is meant by spintronics (it is not meant in the traditional sense, perhaps the authors mean spin-based quantum computing?). Either way, Ref. 4 is not appropriate here, as it has little to do with spin-based applications.

B. Lines 20-22: While accurate, the case is not yet made why we should care.

C. Lines 27-28: need citations here. The single NV linewidth in $[N] \sim 100$ ppm diamond is

typically a few hundred GHz at IHe temperature, so this already indicated the timescales for dynamics probed here. One relevant study in similar samples: Wolters PRL 2013 <http://journals.aps.org/prl/abstract/10.1103/PhysRevLett.110.0274> 01 .

D. Line 40: CI is not a term I am familiar with and the physics of it is never explained in the manuscript.

E. Paragraph beginning with line 55: Because of the use of high-[N] sample, some of these comparisons with work from single NVs in ultra-pure diamond are not fair. Naively I would assume the dynamics in these samples could be more complicated presumably owing to interaction with the many nearby fluctuating charge traps. Need to carefully refute if this is not the case.

F. Line 62: since PA spectroscopy has hardly been explained yet, it is not even clear yet that it is a time domain spectroscopy.

G. Line 67: "ground state bleaching". I did not understand this jargon nor could I really appreciate why being a little to the blue of the ZPL would result in no orbital dynamics.

H. Line 68: " $S_{\text{aniso}}=0.1$ ". Why doesn't it go to zero? Can you please rigorously define what S_{aniso} is and how is it determined from the experimental data? Also, how do we know what we see is strictly due to stimulated emission? What about excited state absorption?

I. Line 68: On the use of biexponentials, I am sometimes suspicious when biexponential decay is used (because why not tri exponential? or some other function?). Please explain exactly what the fit function was and what parameters were fixed and which allowed to vary.

J. Line 77-78: are we saying that $S_{\text{aniso}}=0.1$ is expected in such a doubly-degenerate excited state, singly degenerate ground state? If so, please explain the physics, I do not follow.

K. Lines 81-82: hard to compare since samples were so different

L. Lines 102-103: the statement that one phonon processes don't play a role in orbital dynamics at IHe temperatures is not correct. This is not quite true. See, e.g., Acosta PRL 2012 <http://journals.aps.org/prl/abstract/10.1103/PhysRevLett.108.2064> 01, Goldman PRL 2015 <http://journals.aps.org/prl/abstract/10.1103/PhysRevLett.114.1455> 02

M. Line 145: "unprecedented 5 decades". why is this unprecedented? One would think lots of systems, e.g. dye molecules, QDs, other color centers, etc. could have similar properties?

Reviewer #3 (Remarks to the Author):

Report on "Jahn-Teller-induced femtosecond electronic depolarization dynamics of the nitrogen-vacancy defect in diamond", by Ronald Ulbricht, Shuo Dong, I-Ya Chang, Bala Murali Krishna Mariserla, Keshav M. Dani, Kim Hyeon-Deuk, Zhi-Heng Loh

In this paper the authors study fast dynamic processes in the nitrogen vacancy (NV) centre via femtosecond pump-probe polarization anisotropy. Since the NV centre is of growing importance in terms of quantum science and technology applications it is quite important to understand all aspects of this remarkable system. The authors measure ultra-fast dephasing effects which shed new light on the electronic dynamics of the centre - these results and the emerging picture postulated will be of interest to NV community. My recommendation is for publication, after the authors attend to the following points:

1. This is quite a technical paper and is not well aligned for non-experts, particularly the introduction and abstract. The authors should take more care to explain the jargon they use from the outset, and the physical processes involved.
2. Fig 1 caption could have more detail on the "computed" orbitals with reference to the theoretical methodology.
3. In Fig 2b, the bi-exponential might be more apparent if plotted on a log-scale - perhaps as an inset. This would give the reader a better flow of the definitions of the decay constants the rest of Fig 2. The caption (and in general) could contain more detail than "showing a bi-exponential decay".
4. Fit to data: presumably the result that $\sigma \sim 0$ is due to the flatness of the data at low T. The data looks sufficiently good to quote a value and error from the fit.
5. Referencing needs updating: it seems a bit limited and selective, e.g. thermal imaging is mentioned, but the associated reference (Acosta et al) is not.

Reviewers' comments:

Reviewer #1 (Remarks to the Author):

The manuscript presents ultrafast polarization anisotropy measurements to report electronic dephasing dynamics spanning multiple decades of time associated with the electronic transitions of the nitrogen vacancy defect center in diamond. Additionally, molecular dynamics simulations have been performed to investigate the origin of the observed components.

The paper represents a study that is potentially of significant interest across multiple research communities with implications for future experiments and theoretical development. However, there are some issues with the paper that must be addressed prior to publication as outlined below.

(1.1) The authors invoke a relaxation pathway involving a JT distortion induced conical intersection. While the dynamic JT distortion is certainly associated with non-adiabatic dynamics, the authors need to be cautious when using this terminology. The use of this term requires more explanation and support. For instance, it would be easy for a reader to mistakenly interpret the CI as connecting the excited state to the ground state directly - which would lead to the erroneous idea that relaxation to the ground state occurs on a very short timescale. Also, the term CI is mentioned only at the very beginning of the manuscript and briefly on page 7 associated with fig. 3. At the very least, there should be a more thorough explanation of the role of the CI as presented in fig. 3.

We thank the reviewer for his suggestion to provide an introduction to conical intersections, which would make the article more accessible to the readers. We have added a schematic illustration of a conical intersection as a new figure (Fig. 1b), along with the accompanying caption to briefly explain the features of a conical intersection (REV 1.1). The features of the conical intersection are also reiterated on p. 3 of the revised manuscript (REV 1.1).

In addition, we have also briefly introduced the significance of conical intersections on the same page (REV 1.1): "In the vicinity of CIs, the Born-Oppenheimer approximation breaks down, allowing exceptionally fast NA transitions between potential energy surfaces. In molecular photochemistry, for example, CIs between electronic excited and ground states are known to promote ultrafast internal conversion on femtosecond timescales. In the case of the NV⁻ defect, the CI exists only between the excited state E_x and E_y orbitals, ..."

(1.2) In general, the paper would also benefit from clearer explanations of the results. Although this situation is improved by the supplemental information, there are many aspects of the work presented in the paper that are not sufficiently supported or explained. Even given the length restrictions, this could be done without adding a large amount of text. The paper would be vastly improved by a sentence or two clarifying the main points, results and techniques used. For instance, the vibrational modes contributing to the signal are not listed in a comprehensive manner. Also, no experimental FFT showing the vibrational modes is presented. Only a simulated FFT is shown. This is an issue for claiming the participation of particular JT associated vibrational modes. In addition, the reasons for using polarization anisotropy should be more clearly outlined. This is an appropriate experimental technique to use in this case, however, the authors could do a better job of explaining the motivation to the reader.

This point by the reviewer is well taken. We have included a brief description of the experimental approach on p. 4 (REV 1.2). We have also added a few sentences to the results and discussion section to elaborate on the results. For example, on p. 5 (REV 1.2) we describe the features of the differential transmission spectrum and why the excited-state dynamics appear to the red side of the ZPL transition. This is followed by a description of the implications of a non-vanishing polarization anisotropy.

The reviewer raises an interesting point about the experimental visibility of the computed vibrational modes. First, we note that the modes obtained from AIMD simulations correspond to those of incoherent vibrational motions of the thermal bath. As such, even though such modes can be observed in the simulations, they do not appear in our experiment, since only vibrations that are

coherently excited by the pump pulse would appear as time-domain oscillations in the experimental data. Second, while time-domain oscillations in the PA signal have been observed and attributed to the participation of coherently excited asymmetric vibrational modes (see ref. 22 of the manuscript), our 80-fs time resolution limits the maximum resolvable coherent phonon frequency to ~ 25 meV (assuming that 80-fs corresponds to the half-period of the highest resolvable oscillation frequency). However, the phonon density of states of the NV^- defect vanishes at such low energies (ref. 26). We note that this long pulse duration was chosen to avoid exciting coherent LO phonons that would otherwise dominate the signal (see Fig. S1 of the Supplementary Information) and to suppress excitation of higher vibronic states, e.g., $v' = 1$, thereby reducing interference from vibrational relaxation dynamics.

To address the above, we have added the following on page 7 of the revised manuscript (REV 1.2): “Within the theoretical framework of AIMD, these phonon-induced fluctuations arise from incoherent vibrational motions of the thermal bath, unlike coherent phonons launched by impulsive excitation, which manifest themselves as oscillatory features in the $\Delta T/T$ signals (see Supplementary Information). The FFT power spectrum obtained at 77 K reveals several prominent peaks at 47, 69, 90, 120, 130 and 150-160 meV. These modes have been identified and assigned by earlier ab initio calculations to the various qLVMs of the NV^- defect, including one at 69 meV that coincides with the energy of the JT-active e mode.”

Regarding the motivation to use polarization anisotropy measurements to track the orbital dealignment dynamics, we clarified on page 5 of the revised manuscript (REV 1.2): “ $S_{aniso}(\lambda, t)$ provides information on the alignment dynamics after photoexcitation (see Methods and Supporting Information). In molecular spectroscopy, for instance, this is used to measure the reorientation of molecules in solution, an effect that does not occur here because the NV^- defects are fixed in the diamond lattice. In addition, for probe transitions involving doubly degenerate excited states with perpendicular transition dipoles, as is the case here, the PA signals also reflect electronic reorientation. In such instances, the PA signal reports on electron motion around a CI and its decay yields the dephasing time between the E_x and E_y states.” Important references (refs. 21 – 24) that illustrate these points have also been included in the above.

(1.3) The absorbance seems exceptionally high for an optical experiment. In a third order experiment, absorbances of higher than 0.3 are known to potentially cause issues with contributions from higher order signals. This is true even in third order phase matched directions such as that for the pump probe experiment presented here. This can be mitigated by using lower power pulses (sub 10 nJ as a general guideline) or doing a power dependence to show when higher orders contribute. However, the paper does not list the power per pulse or show a power dependence. Without that information, it is unclear if higher order signals are potentially contributing.

Despite the high static absorbance of the sample, which approaches 0.8 at the ZPL transition due to its narrow linewidth, the normalized differential transmission $\Delta T/T \sim 0.036$ (at the ZPL transition wavelength; see Fig. 1c) remains small due to the low pump fluence (~ 1 mJ/cm²) employed in the experiments. This $\Delta T/T$ value corresponds to a differential absorbance ΔA value of only 0.015. The small $\Delta T/T$ value allows us conclude that higher-order signals do not contribute significantly to our measurement results. We have added this clarification on page 4 of the revised manuscript (REV 1.3): “The typical pump fluence is ~ 1 mJ/cm², which yields a $\Delta T/T$ signal of 0.036 at the ZPL transition wavelength (see Fig. 1c). Such a small $\Delta T/T$ value confirms that our measurements are performed in the weak-perturbative limit and higher-order contributions to the signal are negligible.”

(1.4) The experiments involved polarization anisotropy on a sample with randomly distributed nitrogen vacancy defect centers. In this case, pre-factors associated with a static rotational average should have been considered.

We note that the effects of orientational averaging are already included in our analysis, which follows the seminal work by Wynne and Hochstrasser, cited as ref. 24 in the manuscript. The averages for an isotropic medium are introduced as section 2.5 of that article.

(1.5) Regarding format, the paper needs a conclusion. In the current form, it seems to just end without a statement of significance or a summary of the most important results.

The conclusion is found in the last two paragraphs of the manuscript. To make this clear to the readers, we have added the “Conclusion” heading to that section on p. 10 of the revised manuscript (REV 1.5). In addition, we have also added the “Results and Discussion” heading to the relevant section, starting from p. 4 of the revised manuscript (REV 1.5).

Reviewer #2 (Remarks to the Author):

The article by Ulbricht et al. describes a combined experimental and theoretical investigation of picosecond timescale orbital dynamics within the electronic excited states of the diamond NV center. The methods appear sound (though need to be described a bit more clearly) and the experiments, simulations, and analysis appear to have been performed with a high level of rigor. I am not an expert on the ab initio simulations, so my comments are focused on the experimental and basic theoretical portions which comprise the majority of the manuscript. All in all I recommend that the paper eventually be published in a suitable journal after addressing my comments.

I have several comments on the manuscript which fall under the following categories:

2.1. The methods are not adequately described in the text and there is a reliance on the reader understanding the ultrafast optics jargon.

We thank the reviewer for this comment, which gives us the opportunity to improve the readability of the manuscript. The additional explanations, which will hopefully assist the reader in appreciating the results reported in the manuscript, are listed below.

On page 4, we have added a description of the manner in which time-resolved polarization anisotropy measurements are performed (REV 2.1): “Briefly, the femtosecond PA measurements employ a narrowband linearly polarized pump pulse to excite the ZPL transition of the NV⁻ defect (Fig. 1a), following which a broadband linearly polarized probe pulse measures the pump-induced change of the normalized transmission spectrum $\Delta T/T$. Varying the pump-probe time delay and relative polarization yields the time-resolved $\Delta T/T$ signal for parallel (S^{\parallel}) and perpendicular (S^{\perp}) relative polarization between pump and probe pulses. The polarization anisotropy signal $S_{aniso}(\lambda, t)$ is then obtained from the relation $S_{aniso}(\lambda, t) = (S^{\parallel} - S^{\perp}) / (S^{\parallel} + 2S^{\perp})$ (see Supplementary Information).”

On page 5, we have added an explanation of the prominent features of the normalized differential transmission spectrum, now inserted as a new figure (Fig. 1c). A better understanding of why orbital dynamics appear only on the red side of the ZPL transition will become clearer with this explanation, which reads (REV 2.1): “Photoexcitation leads to increased transmission of the NV⁻ sample, as can be seen from the positive $\Delta T/T$ signal over the entire probe spectrum. Features on the blue side of the ZPL arise from depletion of the 3A_2 ground state by the photoexcitation pump pulse, resulting in the bleaching of the 3A_2 ground state absorption spectrum. The positive $\Delta T/T$ signal on the red side of the ZPL is due to Stokes-shifted stimulated emission from the $v' = 0$ level of the 3E state, populated by the pump pulse, to the various v'' levels on the 3A_2 ground state. As such, the former signal is sensitive to ground state dynamics, whereas the latter is sensitive to excited state dynamics. Note that excited state absorption from the 3E state, which would give negative $\Delta T/T$ signals, is negligible due to the small oscillator strength of excited state absorption into the conduction band.”

2.2. The sample used is a high-defect density [N]~100 ppm sample. These samples have NV centers whose properties at low temperature are dramatically different from the single centers studied in nearly all of the previous works the authors cite. This casts a doubt on the relevance of this work. For example, is it clear the authors are studying processes intrinsic to NV centers and not related to interaction with a rapidly fluctuating charge bath or highly modified phononic environment or some other defect-defect interaction?

The reviewer raises an interesting point regarding the relevance of our work, given the high defect density of the sample. Here, we would like to point out the following observations, which strongly suggest that the ultrafast dynamics reported in the manuscript are inherent to a single defect. First and foremost is the observation that our AIMD simulations reproduce the experimentally measured ultrafast dynamics. With a nitrogen defect density of 100 ppm, about 1 out of 10,000 carbon atoms in the sample is replaced by a nitrogen atom. On the other hand, our AIMD simulations use a supercell in which there is one nitrogen atom for every 215 carbon atoms, i.e., the effective defect density is 4,650 ppm. Even with such a high defect density, the simulations nevertheless reproduce the ~ 0.1 -ps fast component of the biexponential decay for all temperatures, as well as the ~ 10 -ps component of the slow component at 10 K. The excellent agreement between the experimentally measured and theoretically simulated ultrafast dynamics strongly suggest that our results are intrinsic to a single NV⁻ center, free of defect-defect interactions, fluctuations of the charge bath, or a modified phononic environment. Hence, we believe that our results can be compared to the previous measurements cited in the manuscript.

Further related to the issue of a fluctuating charge bath, we are grateful to the reviewer for bringing our attention to the work of Wolters et al. in his comment C below. There, it was shown that the single-defect linewidth could be as large as several hundred GHz, from which the reviewer inferred the existence of ultrafast dynamics reported by us. However, Wolters et al. point out that the few-hundred GHz linewidth results from spectral diffusion (induced by a fluctuating local field) that occurs on the microsecond timescale, orders of magnitude longer than the picosecond and sub-picosecond dynamics measured by us. *More importantly, because this linewidth is limited by spectral diffusion and is not the homogeneous linewidth, it cannot be used to infer dephasing times.* Note that Fu et al. (ref. 12 in the manuscript) had to correct for spectral diffusion in order to determine the homogeneous linewidth of single defects.

Finally, with regards to the effect of a modified phononic environment due to neighboring defects, we further note that previous ab initio simulations by Zhang et al. (ref. 26 in the manuscript), in which a 215-atom supercell was employed, show that the vibrational modes of the NV⁻ defect have a large inverse participation ratio and do not extend to anywhere near the boundaries of the supercell. In other words, the vibrational modes are quasi-localized and the phononic environment of a defect is defined only by the quasi-localized vibrational modes of the defect, with negligible contributions from other defects. This is especially so, given that the defect density is lower for the sample (100 ppm) than the simulations (4,650 ppm).

The above points have been included as a new paragraph on p. 9 – 10 of the revised manuscript (REV 2.2): “We note that previous frequency- and time-domain measurements were performed on high-purity Type 2a diamond samples, which have defect densities that are in the parts-per-billion (ppb) regime, and thus orders of magnitude lower than the Type 1b sample (~ 100 ppm) used in the present study. Therefore a natural question that arises is the extent to which the observed ultrafast dynamics are intrinsic to a single defect. Fluctuations of the charge bath and defect-defect interactions could presumably lead to the enhanced dephasing rates observed herein. First, we note that previous single-defect measurements on samples with similarly high defect densities yield microsecond timescales for spectral diffusion, orders of magnitude longer than the picosecond and sub-picosecond dynamics observed here. Second, considering the quasi-localized nature of the vibrational modes, the modification of the phonon density of states of a given NV⁻ center by an adjacent defect is expected to be negligible. Finally, and most importantly, we point out that the AIMD simulations were performed on a 215-atom supercell, which corresponds to an effective defect density of 4,650 ppm, $>40\times$ larger than the actual ~ 100 ppm defect density of the sample used in the measurements. The fact that the observed dephasing timescales are reproduced by the AIMD simulations strongly suggests that the ultrafast dephasing dynamics are intrinsic to the NV⁻ center. This claim can be verified by future measurements on defects with varying NV⁻ densities.”

2.3. While the experimental techniques are impressive, the topic is a bit esoteric. It's hard to imagine this work impacting how researchers use NV centers in applications. The techniques themselves may be novel, but I am not qualified to judge.

We thank the reviewer for this candid comment. While we agree with the reviewer that our results do not necessarily have a large impact in terms of uncovering new applications of the NV⁻ defect, we nevertheless hope that the reviewer would agree with the importance of our work in elucidating the fundamental photophysics of the NV⁻ center. In particular, our study shows how nonadiabatic dynamics at the Jahn-Teller conical intersection can drive ultrafast orbital dealignment dynamics on the ~100-fs time scale. In fact, the precise nature and influence of the JT distortion is one of the few remaining fundamental problems in the physics of the NV⁻ defect, a fact that was recently recognized in the most comprehensive review on its properties (ref. 1 in the manuscript).

Specific comments below:

A. Line 19: Not clear what is meant by spintronics (it is not meant in the traditional sense, perhaps the authors mean-spin-based quantum computing?). Either way, Ref. 4 is not appropriate here, as it has little to do with spin-based applications.

We thank the reviewer for pointing this out. We have changed “spintronics” to “spin-based quantum computing” (REV 2.A) and removed reference 4.

B. Lines 20-22: While accurate, the case is not yet made why we should care.

We have modified the abstract to emphasize the hitherto unresolved ultrafast orbital averaging dynamics (REV 2.B).

C. Lines 27-28: need citations here. The single NV linewidth in [N]~100 ppm diamond is typically a few hundred GHz at lHe temperature, so this already indicated the timescales for dynamics probed here. One relevant study in similar samples: Wolters PRL 2013 <http://journals.aps.org/prl/abstract/10.1103/PhysRevLett.110.027401> .

We thank the reviewer for bringing our attention to the nice work by Wolters et al. In response to this comment, we would like to direct the attention of the reviewer to the second paragraph of our response to the earlier comment 2.2. Briefly, we note that the linewidth of a few hundred GHz reported by Wolters et al. has not been corrected for spectral diffusion, and therefore does not represent a homogeneous linewidth that can be used to infer the dephasing time. Note that Fu et al. (ref. 12 in the manuscript) had to correct for spectral diffusion in order to determine the homogeneous linewidth of single defects.

D. Line 40: CI is not a term I am familiar with and the physics of it is never explained in the manuscript.

This comment is related to comment 1.1 by reviewer #1. Hence, we would like to direct the attention of the reviewer to our response to the earlier comment. Briefly, we have added a figure (Fig. 1b) and explanatory text to introduce the features and the significance of conical intersections.

E. Paragraph beginning with line 55: Because of the use of high-[N] sample, some of these comparisons with work from single NVs in ultra-pure diamond are not fair. Naively I would assume the dynamics in these samples could be more complicated presumably owing to interaction with the many nearby fluctuating charge traps. Need to carefully refute if this is not the case.

This comment is related to comment 2.2 by this reviewer. We refer the reviewer to our response to comment 2.2 above.

F. Line 62: since PA spectroscopy has hardly been explained yet, it is not even clear yet that it is a time domain spectroscopy.

This comment is related to comment 2.1 by this reviewer. We state in the second paragraph of our response to the earlier comment 2.1 that we have included in the revised manuscript a more detailed description of polarization anisotropy spectroscopy.

G. Line 67: "ground state bleaching". I did not understand this jargon nor could I really appreciate why being a little to the blue of the ZPL would result in no orbital dynamics.

This comment is related to comment 2.1 by this reviewer. We state in the third paragraph of our response to the earlier comment 2.1 that we have included in the revised manuscript a description of the features of the normalized differential transmission spectrum, and a brief explanation as to why the spectral features to the blue of the ZPL transition do not encode orbital dynamics. In addition, we have added the differential transmission spectrum as a new figure (Fig. 1c).

H. Line 68: " $S_{\text{aniso}}=0.1$ ". Why doesn't it go to zero? Can you please rigorously define what S_{aniso} is and how is it determined from the experimental data? Also, how do we know what we see is strictly due to stimulated emission? What about excited state absorption?

We thank the reviewer for this question, which we note is related to comments 2.1 and 2.F of this reviewer. We have now provided the definition of S_{aniso} in the second paragraph on p. 4 of the revised manuscript: $S_{\text{aniso}}(\lambda, t) = (S^{\parallel} - S^{\perp}) / (S^{\parallel} + 2S^{\perp})$. A brief description of how S_{aniso} is determined is given in the same paragraph and more details can be found on p. 7–8 of the Supplementary Information.

The seminal work by Wynne and Hochstrasser (ref. 24 in the manuscript) shows that the anisotropy approaches 0.1 upon complete electronic dephasing of doubly degenerate states (see eq. 22 in that article). S_{aniso} would vanish only if the system of interest could undergo free rotation, as in the case of molecules in solution (however, note that the effect of rotation is neglected in ref. 24 because it occurs on timescales much longer than electronic dephasing). On the other hand, in the case of the NV^- defects, the defects are covalently fixed relative to the lattice, i.e., they are not free to rotate. Hence, S_{aniso} cannot completely vanish. This is clarified on p. 6 of the revised manuscript (REV 2.H): "We note that S_{aniso} does not vanish to zero, as it would for molecules in solution by rotational diffusion, because the defects are fixed in the diamond lattice."

Regarding the possible contribution from excited-state absorption to S_{aniso} , we note that the differential transmission spectrum shows a positive $\Delta T/T$ signal over the entire probe spectrum. Because stimulated emission results in $\Delta T/T > 0$ whereas excited-state absorption results in $\Delta T/T < 0$, we conclude that there is negligible contribution from the latter to the signal. This is emphasized in the first paragraph on p. 5 (REV 2.H): "Note that excited state absorption from the 3E state, which would give negative $\Delta T/T$ signals, is negligible due to the small oscillator strength of excited state absorption into the conduction band."

I. Line 68: On the use of biexponentials, I am sometimes suspicious when biexponential decay is used (because why not tri exponential? or some other function?). Please explain exactly what the fit function was and what parameters were fixed and which allowed to vary.

Measurements performed over a temperature range of 10 – 300 K reveal pronounced variations of the anisotropy decay behavior over this broad temperature range, from which we can conclude that the temperature-dependent anisotropy shows two decay components. Introducing additional components would over-parameterize the fit and result in non-unique decay constants. One might consider functions other than exponentials that can be used to describe decays, e.g., Gaussian decays emerge as a result of non-Markovian dynamics. Nevertheless, exponential decays, which arise from transition rates that obey the Fermi Golden Rule, are by far the most common in ultrafast phenomena. Note that previous studies of anisotropy decays, such as the seminal work by Wynne and Hochstrasser (ref. 24 in the manuscript), also employ exponential functions in their analysis. Moreover, the signal-to-noise of our measurements does not allow subtle differences in decay shapes to be distinguished. The above considerations lead us to describe the measured dynamics as biexponential decays.

We note that all the measured PA decays could be fit to a biexponential decay model, which we believe lends credibility to our approach. At low temperatures, both time constants differ by up to two orders of magnitude (100 fs vs. 10 ps) and are clearly distinguishable. The slow component quickly decreases with increasing temperature until it becomes comparable to the fast component at 150 K (100 fs vs. 700 fs). Above this temperature both components cannot be distinguished anymore, and thus our fit becomes single-exponential.

We apologize for not showing the fit function, which is now stated explicitly on p. 6 of the revised manuscript (REV 2.I): “The PA decay can be fit to the function $S_{aniso}(t) = A_0 + A_1 e^{-t/\tau_1} + A_2 e^{-t/\tau_2}$, where the offset A_0 , amplitudes A_1 and A_2 , and exponential decay constants τ_1 and τ_2 are all fit parameters.”

J. Line 77-78: are we saying that $S_{aniso}=0.1$ is expected in such a doubly-degenerate excited state, singly degenerate ground state? If so, please explain the physics, I do not follow.

This comment is related to comment 2.H by this reviewer. We refer the reviewer to our response to comment 2.H above. Here, we can confirm that a system with a doubly degenerate excited state and a singly degenerate ground state should indeed yield an asymptotic S_{aniso} value of 0.1. Such a system is exactly identical to the case that was first treated theoretically by Wynne and Hochstrasser (ref. 24) and subsequently experimentally verified by the same group [*Chem. Phys. Lett.* **206**, 493–499 (1996)] and by others (refs. 22 & 23). The reviewer is kindly referred to ref. 24 for a thorough mathematical derivation and a detailed explanation of the physical phenomena that underlie the asymptotic S_{aniso} value.

K. Lines 81-82: hard to compare since samples were so different

This comment is related to comment 2.2 by this reviewer. We refer the reviewer to our response to comment 2.2 above.

L. Lines 102-103: the statement that one phonon processes don't play a role in orbital dynamics at IHe temperatures is not correct. This is not quite true. See, e.g., Acosta PRL 2012 <http://journals.aps.org/prl/abstract/10.1103/PhysRevLett.108.206401>, Goldman PRL 2015 <http://journals.aps.org/prl/abstract/10.1103/PhysRevLett.114.145502>

We thank the reviewer for bringing these articles to our attention, which we have now added as refs. 28 (Goldman) and 32 (Acosta) in the revised manuscript. However, we did not find in these articles any concrete evidence for one-phonon processes being significant at low temperatures, contrary to what the reviewer wrote. In Acosta et al., it is only stated that mixing between E_x and E_y is “maybe due to a single-phonon orbital relaxation process” and that “a detailed study will be the focus of future work.” Indeed, in the absence of a temperature-dependent study, it would have been difficult for Acosta et al. to ascribe the observed orbital mixing to a one-phonon process (or for the matter, any phonon-related process). In Goldman et al. it is stated explicitly that “There also exists a one-phonon emission process whose contribution to the mixing rate scales as $\Delta_{xy}^2 T$. This contribution is negligible in our experiment because of the small density of states for phonons of $\Delta_{xy} = 3.9$ GHz.” This is precisely what we have stated in the manuscript. Moreover, we note that the small phonon density of states at such low frequencies is the reason Fu et al. (ref. 12) had to consider a two-phonon Raman-type population transfer in explaining their temperature-dependent linewidth data.

M. Line 145: “unprecedented 5 decades”. Why is this unprecedented? One would think lots of systems, e.g. dye molecules, QDs, other color centers, etc. could have similar properties?

We believe that there might have been a misunderstanding because we did not phrase our original statement clearly. We do not mean to say that it is unprecedented for dephasing times to span several orders of magnitude over a large temperature range, i.e., from cryogenic to room temperature. This indeed happens for a multitude of systems, as pointed out by the reviewer. Instead, we mean to say that *at a given cryogenic temperature*, say, 10 K, various electronic dephasing processes of the NV^- defect are found to occur on timescales that differ by up to five orders of magnitude, from ~ 100 fs for the nonadiabatic transition as measured in our experiments to ~ 10 ns as inferred from frequency-domain linewidth measurements. We are not aware of other systems that exhibit such behavior. We have revised the last sentence of the concluding paragraph to clarify this point (REV 2.M). The sentence now reads, “In the case of the NV^- center, our results, together with earlier frequency and time-domain studies, demonstrate that electronic dephasing at a given cryogenic temperature spans an unprecedented five decades in time, from 10^{-13} to 10^{-8} s.”

There have been reports of bi-phasic dephasing of excitons in self-assembled and colloidal quantum dots (refs. 34 & 35 in the manuscript) that span three orders of magnitude in time scales, from ~ 1 ps

to ~1 ns. We have pointed this out on p. 10 – 11 of the revised manuscript (REV 2.M): “Note that time-domain photon echo measurements have elucidated similar biphasic dephasing dynamics spanning two to three orders of magnitude for excitons in self-assembled and colloidal quantum dots.” Nevertheless, we emphasize that the five-orders of magnitude span in dephasing times exhibited by NV⁻ defect remains unsurpassed.

Reviewer #3 (Remarks to the Author):

Report on "Jahn-Teller-induced femtosecond electronic depolarization dynamics of the nitrogen-vacancy defect in diamond", by Ronald Ulbricht, Shuo Dong, I-Ya Chang, Bala Murali Krishna Mariserla, Keshav M. Dani, Kim Hyeon-Deuk, Zhi-Heng Loh

In this paper the authors study fast dynamic processes in the nitrogen vacancy (NV) centre via femtosecond pump-probe polarization anisotropy. Since the NV centre is of growing importance in terms of quantum science and technology applications it is quite important to understand all aspects of this remarkable system. The authors measure ultra-fast dephasing effects which shed new light on the electronic dynamics of the centre - these results and the emerging picture postulated will be of interest to NV community. My recommendation is for publication, after the authors attend to the following points:

3.1. This is quite a technical paper and is not well aligned for non-experts, particularly the introduction and abstract. The authors should take more care to explain the jargon they use from the outset, and the physical processes involved.

We thank the reviewer for this comment, which gives us the opportunity to improve the readability of the manuscript. This comment is also similar to comments 1.1 and 2.1 by the first two reviewers. We refer the reviewer to our responses to comments 1.1 and 2.1 above. Briefly, we have provided a condensed description of (1) features and significance of conical intersections, (2) features present in the differential transmission spectrum and what they mean, and (3) time-resolved polarization anisotropy spectroscopy and how the data is collected and processed. We believe that these additional explanations make it easier for non-expert readers to appreciate the article. In addition, we have also revised the abstract to state clearly the importance of our work.

3.2. Fig 1 caption could have more detail on the "computed" orbitals with reference to the theoretical methodology.

We have inserted into the Fig. 1 caption (REV 3.2): “The plots are calculated by density functional theory using the PBE functional and projector-augmented-wave pseudopotentials (see Methods).”

3.3. In Fig 2b, the bi-exponential might be more apparent if plotted on a log-scale - perhaps as an inset. This would give the reader a better flow of the definitions of the decay constants the rest of Fig 2. The caption (and in general) could contain more detail than "showing a bi-exponential decay".

A log scale would indeed be a good way to visualize the two components of the bi-exponential decay. However, due to the y-offset of the curve, whose value is dictated by the asymptotic S_{aniso} value of 0.1, the decay essentially only involves a decrease of the anisotropy by a factor of less than two. As a result, the log plot, shown in the left graph below, is hardly different from a linear plot. Only if the curve approaches zero would the decay of the signal span several orders of magnitude, and therefore clearly show linear slopes in a log plot. One way to achieve this is by subtracting the y-offset so that the anisotropy decays to zero, as shown in the right graph. Introducing such an offset, however, might appear somewhat confusing to the reader. We hope the referee agrees with us.

3.4. Fit to data: presumably the result that $\sigma \sim 0$ is due to the flatness of the data at low T. The data looks sufficiently good to quote a value and error from the fit.

The fit to the data gives $\sigma = (0.5 \pm 5.7) \times 10^{-4} \text{ ps}^{-1}\text{K}^{-1}$. Because the error is much larger than the value of the fit value, we conclude that $\sigma \sim 0$ to within experimental uncertainty.

3.5. Referencing needs updating: it seems a bit limited and selective, e.g. thermal imaging is mentioned, but the associated reference (Acosta et al) is not.

We thank the reviewer for this comment. We have updated the references to provide a more comprehensive coverage of the various novel applications of the NV^- defect (refs. 6, 7, 9–11 are new), including the work of Acosta et al. on thermal imaging (ref. 9).

REVIEWERS' COMMENTS:

Reviewer #2 (Remarks to the Author):

The authors have done a thorough job responding to my comments, and the article is now on stronger technical footing.

I still have some questions about impact/novelty, but I believe those can best be addressed by experts in the ultra-fast community and the journal editors.

Reviewer #3 (Remarks to the Author):

In the revision and (extensive) rebuttal the authors have generally addressed the referees' technical issues and questions. Overall, my recommendation is for publication, subject to the following editorial matter. Presentation issues raised by the referees have been addressed to some extent, primarily through a much-improved Fig 1, however, the abstract and intro read essentially the same – very heavy on jargon (one sentence alone in the abstract defines four abbreviations – PA, AIMD, JT, CI) and would be generally difficult to access, even for an expert. To greatly improve readability it would not take more than a line or two in the introduction at appropriate junctures to actually explain the physical effects that are central to the paper's results and conclusions.

Reviewers' comments

Reviewer #2 (Remarks to the Author):

The authors have done a thorough job responding to my comments, and the article is now on stronger technical footing.

I still have some questions about impact/novelty, but I believe those can best be addressed by experts in the ultra-fast community and the journal editors.

We appreciate the Reviewer's concern regarding the impact/novelty of our work. Our work represents one of the earliest reports on the application of ultrafast spectroscopy to elucidate the electronic dynamics of the NV⁻ defect in diamond, complementing an earlier study that employed femtosecond two-dimensional electronic spectroscopy to identify the vibrational modes that are coupled to the optical transition of the NV⁻ defect [Fleming et al., *Nature Phys.* **9**, 744–749 (2013)]. Our results shed light on the origin of the loss of polarization fidelity in the photoluminescence of the NV⁻ defect, which in turn limits its application as a single-photon quantum emitter. Nonadiabatic transitions that are found to drive electronic depolarization are generic to other solid-state defects containing doubly degenerate electronic excited states that are also candidates for quantum technology applications, such as the silicon-vacancy in diamond and the divacancy in silicon carbide. For these reasons, we believe that our work will be of interest to readers in the ultrafast spectroscopy and NV⁻ defect communities.

Reviewer #3 (Remarks to the Author):

In the revision and (extensive) rebuttal the authors have generally addressed the referees technical issues and questions. Overall, my recommendation is for publication, subject to the following editorial matter. Presentation issues raised by the referees have been addressed to some extent, primarily through a much-improved Fig 1, however, the abstract and intro read essentially the same – very heavy on jargon (one sentence alone in the abstract defines four abbreviations – PA, AIMD, JT, CI) and would be generally difficult to access, even for an expert. To greatly improve readability it would not take more than a line or two in the introduction at appropriate junctures to actually explain the physical effects that are central to the papers results and conclusions.

We are grateful to the Reviewer for recommending our work for publication. We have revised the abstract thoroughly in order to improve its readability. In particular, we have taken the jargon-heavy sentence that contains four abbreviations (PA, AIMD, JT, and CI) and spread its key points over two sentences: 1) Femtosecond polarization anisotropy (PA) spectroscopy is employed to interrogate electronic dynamics, 2) Ab initio molecular dynamics (AIMD) is used to explain the experimentally observed ultrafast depolarization timescales. The Jahn-Teller (JT) distortion and its associated conical intersection (CI) are mentioned only in the Introduction section. In addition, we have more clearly stated in the abstract the motivation for our work – that the NV⁻ defect, despite being a prime candidate for serving as a single-photon quantum emitter, suffers from the hitherto unexplained loss of polarization fidelity in its photoluminescence even with polarized light excitation. Aside from the amendments to the abstract, we have also modified the 4th paragraph of the introduction section to explicitly state that the ultrafast nonadiabatic transitions observed in our work arise from the conical intersection in the ³E excited state: “In the case of the NV⁻ defect, the CI exists only between the excited state E_x and E_y orbitals, **thereby potentially enabling ultrafast nonadiabatic transitions between them.**”(inserted text in bold) This piece of information, critical to the understanding of the physical phenomena uncovered in our experiments and simulations, was not stated explicitly in earlier versions of the manuscript. We sincerely apologize for this oversight. We hope that the Reviewer would find the above amendments to the abstract and the introduction satisfactory.